# Anomaly Detection with Variance Stabilized Density Estimation

**Amit Rozner**[1*]     **Barak Battash**[1*]     **Henry Li**[2]     **Lior Wolf**[3]     **Ofir Lindenbaum**[1]

[1]Faculty of Engineering, Bar Ilan University, Ramat-Gan, Israel
[2]Department of Applied Mathematics, Yale University, New Haven, Connecticut
[3]School of Computer Science, Tel Aviv University, Tel-Aviv, Israel
[*]These authors contributed equally

## Abstract

We propose a modified density estimation problem that is highly effective for detecting anomalies in tabular data. Our approach assumes that the density function is relatively stable (with lower variance) around normal samples. We have verified this hypothesis empirically using a wide range of real-world data. Then, we present a *variance-stabilized density estimation* problem for maximizing the likelihood of the observed samples while minimizing the variance of the density around normal samples. To obtain a reliable anomaly detector, we introduce a spectral ensemble of autoregressive models for learning the *variance-stabilized* distribution. We have conducted an extensive benchmark with 52 datasets, demonstrating that our method leads to state-of-the-art results while alleviating the need for data-specific hyperparameter tuning. Finally, we have used an ablation study to demonstrate the importance of each of the proposed components, followed by a stability analysis evaluating the robustness of our model.

## 1 INTRODUCTION

Anomaly detection (AD) is a crucial task in machine learning. It involves identifying patterns or behaviors that deviate from what is considered normal in a given dataset. Accurate identification of anomalous samples is essential for the success of various applications such as fraud detection [Hilal et al., 2021], medical diagnosis [Fernando et al., 2021, Irshaid et al., 2022, Farhadian et al., 2022], manufacturing [Liu et al., 2018], explosion detection [Rabin et al., 2016, Bregman et al., 2021] and more.

An intuitive and well-studied perspective on anomaly detection is via the lens of density estimation. In this method, a probabilistic model is trained to maximize the average log-likelihood of non-anomalous or "normal" samples. Any sample that has a low likelihood value under the learned density function is considered anomalous. For instance, Histogram-based Outlier Score (HBOS) [Goldstein and Dengel, 2012] uses a histogram of features to score anomalies. Variational autoencoders [An and Cho, 2015] use a Gaussian prior for estimating the likelihood of the observations. The Copula-Based Outlier Detection method (COPOD) [Li et al., 2020] models the data using an empirical copula and identifies anomalies as "extreme" points based on the left and right tails of the cumulative distribution function.

While the low-likelihood assumption for modeling anomalous samples seems realistic, density-based anomaly detection methods often underperform compared to geometric or one-class classification models [Han et al., 2022]. This gap has been explained by several authors. One possible explanation is the challenge of density estimation due to the curse of dimensionality, which often leads to overfitting [Nalisnick et al., 2019, Wang and Scott, 2019, Nachman and Shih, 2020]. Another argument is that even "simple" examples may result in high likelihoods, despite not being seen during training [Choi et al., 2018, Nalisnick et al., 2019]. To bridge this gap, we propose a regularized density estimation problem that prevents overfitting and significantly improves the ability to distinguish between normal and abnormal samples.

We base our work on a new assumption on the properties of the density function around normal samples. Specifically, our key idea is to model the density function of normal samples as roughly uniform in a compact domain, which results in a more stable density function around inliers as compared to outliers. We first provide empirical evidence to support our stable density assumption. We then propose a variance-stabilized density estimation (VSDE) problem that is realized as a regularized maximum likelihood problem. We propose a spectral ensemble of multiple autoregressive models implemented using probabilistically normalized networks (PNNs) to learn a reliable and stable density estimate. Each model is trained to learn a density representation

of normal samples that is uniform in a compact domain. We have conducted an extensive benchmark with 52 real-world datasets, which demonstrates that our approach is a new state-of-the-art anomaly detector for tabular data. A schematic illustration of this procedure appears in Figure 1.

## 2 RELATED WORK

One popular line of solutions for AD relies on the geometric structure of the data. These include methods such as Local Outlier Factor (LOF) [Breunig et al., 2000], which locates anomalous data by measuring local deviations between the data point and its neighbors. Another method is to use the distance to the $k$ nearest neighbors ($k$-NN) to detect anomalies. AutoEncoder (AE) can also be used to detect anomalies by modeling them as harder-to-reconstruct samples [Zhou and Paffenroth, 2017]. This approach was later improved by Chen et al. [2017], who presented an ensemble of AE with different dropout connections. Recently, Lindenbaum et al. [2021] presented a robust AE that can exclude anomalies during training using probabilistic gates.

One-class classification is a well-studied paradigm for detecting anomalies. Deep One-Class Classification [Ruff et al., 2018] trains a deep neural network to learn a transformation that minimizes the volume of a hypersphere surrounding a fixed point. The distance of a sample from the center of the hypersphere is used to detect anomalies. Self-supervision has been used in several studies to enhance the classifier's ability to distinguish between normal and abnormal samples. For example, Qiu et al. [2021] applies affine transformations to non-image datasets and uses the likelihood of a contrastive predictor to detect anomalies. Another approach is Internal Contrastive Learning (ICL) presented by Shenkar and Wolf [2022], which uses a special masking scheme to learn an informative anomaly score.

Density-based anomaly detection is a technique that works under the assumption that anomalous events occur rarely and are unlikely. Therefore, a sample that is unlikely is considered to have low "likelihood" and high probability density for a normal sample. Several studies have followed this intuition, implicitly or explicitly [Liu et al., 2020, Bishop, 1994, Hendrycks et al., 2018], even in classification [Chalapathy et al., 2018, Ruff et al., 2018, Bergman and Hoshen, 2020, Qiu et al., 2021] or reconstruction [Chen et al., 2018, 2017] based anomaly detection. However, recent research has pointed out that anomaly detection based on simple density estimation has several flaws. According to Le Lan and Dinh [2021], methods based on likelihood scoring are unreliable even when provided with a perfect density model of in-distribution data. Nalisnick et al. [2019] demonstrated that the regions of high likelihood in a probability distribution may not be associated with regions of high probability, especially as the number of dimensions increases. Furthermore, Caterini and Loaiza-Ganem [2022] focuses on the impact of the entropy term in anomaly detection and suggests looking for lower-entropy data representations before performing likelihood-based anomaly detection.

According to Li et al. [2022], non-parametric learning of data distribution can be achieved by utilizing the empirical cumulative distribution per data dimension. They aggregate the estimated tail probabilities across dimensions to compute an anomaly score. Livernoche et al. [2023] and Zamberg et al. [2023] have recently shown that diffusion models have potential in anomaly detection. Livernoche et al. [2023] present Diffusion Time Estimation (DTE) that approximates the distribution over diffusion time for each input, and use those as anomaly scores. In this work, we present a new *variance-stabilized density estimation* problem and show that it is highly effective for detecting anomalies in tabular data.

## 3 METHOD

**Problem Definition**  Given samples $X = \{x_1, \ldots, x_N\}$, where $x_i \in \mathbb{R}^D$, we model the data by $X = X_N \cup X_A$, where $X_N$ are normal samples and $X_A$ are anomalies. Our goal is to learn a score function $S : \mathbb{R}^D \to \mathbb{R}$, such that $S(x_n) > S(x_a)$, for all $x_n \in X_N$ and $x_a \in X_A$ while training solely on $x \in X_N$. In this study, we consider the modeling of $S()$ by estimating a regularized density of the normal samples.

**Intuition**  It is commonly assumed in the anomaly detection field that normal data has a simple structure, while anomalies do not follow a clear pattern and can be caused by many unknown factors [Ahmed et al., 2016]. Density-based models for anomaly detection [Bishop, 1994] work by assuming that the density of the data $p_X(\cdot)$ is typically higher for normal samples than for anomalies ($p_X(x_n) > p_X(x_a)$ for $x_n \in X_N$ and $x_a \in X_A$). However, recent research has shown that relying solely on the likelihood score of a density model is not always effective (as discussed in Sec. 2). To address this issue, one potential solution is to apply regularization to the density estimation problem, which can reduce overfitting [Rothfuss et al., 2019].

In Legaria et al. [2023], the authors show that different anomaly detection distance metrics perform better for uniformly distributed normal data. Their approach inspired us to introduce a new assumption for modeling the density function of normal samples. Specifically, our working hypothesis is that the density function around normal samples is stable, which means that the variance of $\log p_X(x)$ is relatively low. This hypothesis holds true if the normal samples come from a uniform distribution. Our main idea is to regularize the density estimation problem by ensuring that the estimated density has a low variance around normal samples.

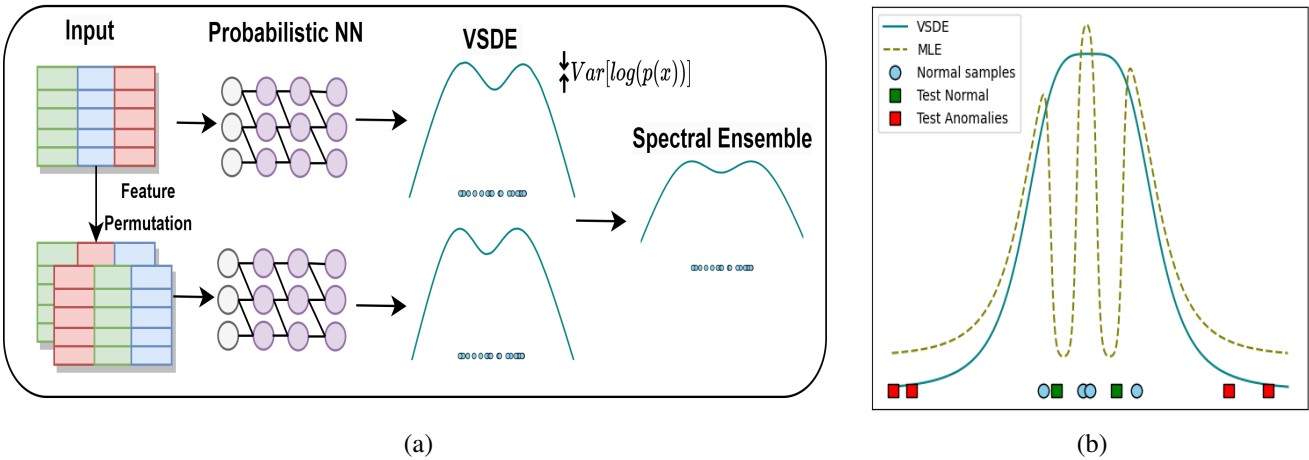

(a)                                      (b)

Figure 1: (a) The proposed framework for anomaly detection. Our method involves using multiple versions of permuted tabular data, which are fed into a Probabilistic Normalized Network (PNN). The PNN is designed to model the density of normal samples as uniform in a compact domain. Each PNN is trained to minimize a regularized negative log-likelihood loss (see Eq. 1). Since our PNN is implemented using an autoregressive model, we use a spectral ensemble of the learned log-likelihood functions as an anomaly score for unseen samples. (b) Illustration of the proposed variance-stabilized density estimation (VSDE) vs. standard (un-regularized) maximum likelihood estimation (MLE) for one-dimensional data. During training, the VSDE learns a more "stable" density estimate around normal samples. This results in a better likelihood estimate for distinguishing between normal and abnormal samples at test time. These findings are supported empirically by our experimental results.

**Empirical Evidence** We evaluated our low variance assumption using a diverse set of 52 publicly available tabular anomaly detection datasets. For each dataset, we estimated the variance of the log-likelihood of normal and anomalous samples. To do this, we calculated $\hat{\sigma}_n^2 = \mathbb{E}_{X_N}(\log \hat{p}_\theta(x) - \mu_n)^2$ for normal samples and $\hat{\sigma}_a^2 = \mathbb{E}_{X_A}(\log \hat{p}_\theta(x) - \mu_a)^2$ for anomalous samples, where $\mu_n$ and $\mu_a$ are the means of the log density estimated over the normal and anomalous samples, respectively.

We visualized the log-likelihood variance ratio between normal and anomalous samples in Figure 2. To compute the variance ratio, we addressed the imbalance between normal and anomalous samples by randomly selecting normal samples to match the quantity of anomalous ones. As indicated by this figure, in most of the datasets (47 out of 52), the variance ratio is smaller than 1 (below the dashed line). This supports our hypothesis that the density around normal samples has a relatively low variance. In [Ye et al., 2023], the authors provide related empirical evidence that multiple classifiers trained on normal samples have lower variance than those trained on anomalous samples. We exploited our assumption to derive a modified density estimation for learning a stabilized density of normal samples.

**Regularized density estimation** Following recent anomaly detection works [Bergman and Hoshen, 2020, Qiu et al., 2021, Shenkar and Wolf, 2022], during training, we only assume access to normal samples, $\mathcal{X}_{train} \subset X_N$. To incorporate our low variance assumption, we have

formulated a modified density estimation problem that imposes stability of the density function. To achieve this, we minimize a regularized version of the negative log-likelihood. Denoting a density estimator that is parameterized by $\theta$ as $\hat{p}_\theta(x)$, our optimization problem can be written as:

$$\min -\mathbb{E}_{X_N}\big[\log \hat{p}_\theta(x)\big] + \lambda \text{Var}_{X_N}\big[\log \hat{p}_\theta(x)\big], \quad (1)$$

where $\lambda$ is a hyperpramater that controls the amount of regularization. Specifically, for $\lambda = 0$, Eq. 1 boils down to a standard maximum likelihood problem, and using larger values of $\lambda$ encourages a more stable (lower variance) density estimate.

In recent years, various deep-learning techniques have been proposed for density estimation. Among them, we have chosen an autoregressive model to learn $\hat{p}_\theta(x)$, as it has shown superior performance on density estimation benchmarks. Although normalizing flow based models are also a well-studied alternative [Dinh et al., 2014, 2016, Kingma et al., 2016, Meng et al., 2022], we opted for the autoregressive probabilistic model for their simplicity. The likelihood of a sample $x \in \mathcal{X}_{train}$ is expressed based on this model.

$$\hat{p}_\theta(x) = \prod_{i=1}^{D} \hat{p}_\theta(x^{(i)}|x^{(<i)}),$$

$$\log \hat{p}_\theta(x) = \sum_{i=1}^{D} \log \hat{p}_\theta(x^{(i)}|x^{(<i)}), \quad (2)$$

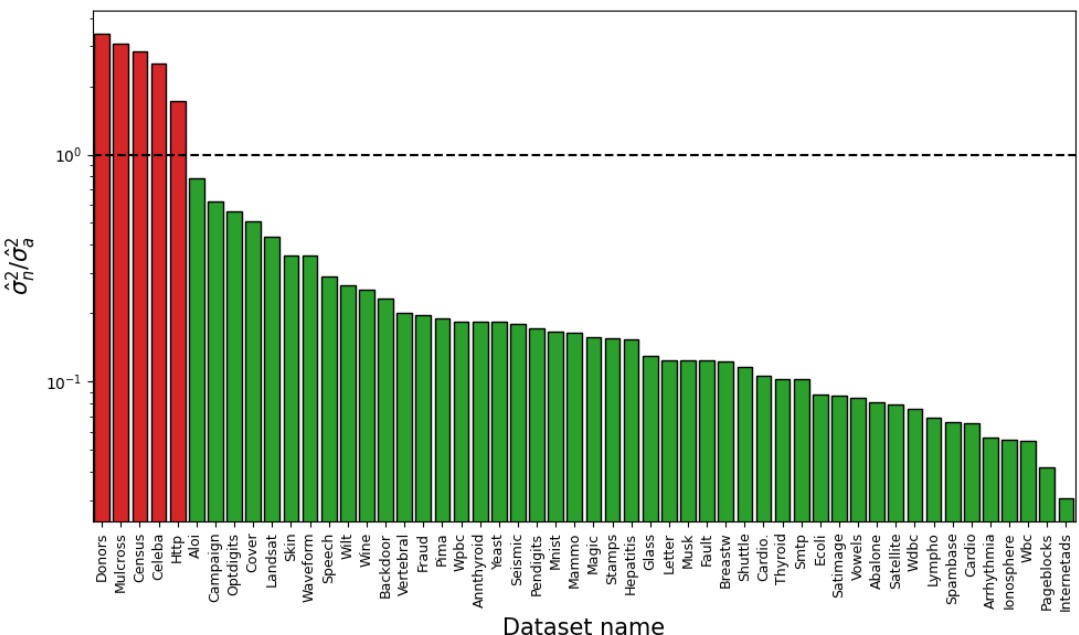

Figure 2: Evaluation of our "stable" density assumption. We plot the mean log-likelihood variance ratio between normal and anomalous samples (see definition in the Intuition section below) for 52 publicly available tabular datasets. Values above the dashed line are greater than 1. Our results indicate that in most datasets, the density is more stable (with lower variance) around normal samples than anomalies. This corroborates our assumptions and motivates our proposed modified density estimation problem for anomaly detection.

where $x^{(i)}$ is the $i$-th feature of $x$, and $D$ is the input dimension. To alleviate the influence of variable order on our estimate, we present below a new type of spectral ensemble of likelihood estimates, each based on a different permutation of features.

To estimate our stabilized density, we harness a recently proposed probabilistic normalized network (PNN) [Li and Kluger, 2022]. Assuming the density of any feature $x^{(i)}$ is compactly supported on $[A, B] \in \mathbb{R}$, we can define the cumulative distribution function (CDF) of an arbitrary density as

$$\hat{P}(X^{(i)} \leq x^{(i)}) = \frac{F_\theta(x^{(i)}) - F_\theta(A)}{F_\theta(B) - F_\theta(A)}, \qquad (3)$$

where $F_\theta$ is an arbitrary neural network function with strictly positive weights $\theta$, and is thus monotonic. Since each strictly monotonic CDF is uniquely mapped to a corresponding density, we now have unfettered access to the class of all densities on $[A, B] \in \mathbb{R}$, up to the expressiveness of $F_\theta$ via the relation

$$\hat{p}_\theta(x^{(i)}) = \nabla_x^{(i)} \hat{P}(X \leq x^{(i)}). \qquad (4)$$

By conditioning each $F_\theta(x^{(i)})$ on $x^{(<i)}$, we obtain in their product an autoregressive density on $x$. This formulation enjoys much greater flexibility than other density estimation models in the literature, such as flow-based models [Dinh et al., 2014, 2016, Ho et al., 2019, Durkan et al., 2019] or

even other autoregressive models [Uria et al., 2013, Salimans et al., 2017] that model $x^{(i)}$ using simple distributions (e.g., mixtures of Gaussian, Logistic). Our $\hat{p}_\theta$ represented by $F_\theta$ is provably a universal approximator for arbitrary compact densities on $\mathbb{R}^D$ [Li and Kluger, 2022], and therefore more expressive while still being end-to-end differentiable. The model $F_\theta$ is composed of $n$ layers defined recursively by the relation

$$a_l = \psi(h_A(A_l)^T a_{l-1} + h_b(b_l, A_l)) \qquad (5)$$

where $l$ layer index of the PNN, $a_0 := x$, $\psi$ is the sigmoid activation, and $A_l, b_l$ are the weights and biases of the $l$th layer. The final layer is defined as $F_\theta(x) = softmax(A_n^T a_{n-1})$.

**Feature permutation ensemble**  Our density estimator is autoregressive and it means that different input feature permutations can result in different density estimates (as shown in Eq. 2). However, we have used this characteristic to our advantage and developed an ensemble-based approach for density estimation that is based on randomized permutations of the features, which makes our estimate more robust. To achieve this, we have defined $\mathcal{P}_D$ as the set of permutation matrices of size $D$.

We learn an ensemble of regularized estimators, each minimizing objective Eq. 1 under a different random realization of feature permutation $\Pi_\ell \in \mathcal{P}_D$. We denote by $S(x) = \log \hat{p}_\theta(x)$ as the estimated log-likelihood of $x$. Next,

we compute the score for each permutation, namely $S_\ell(x)$ is the score computed based on the permutation matrix $\Pi_\ell, \ell = 1, ..., N_{perm}$. Finally, inspired by the supervised ensemble proposed in Jaffe et al. [2015], we present a spectral ensemble approach proposed for aggregating multiple density estimation functions.

The idea is to compute the $N_{perm} \times N_{perm}$ sample covariance matrix of multiple log-likelihood estimates

$$\Sigma = \mathbb{E}[(S_i(x) - \mu_i)(S_j(x) - \mu_j)],$$

with $\mu_i = \mathbb{E}(S_i(x))$. Then, utilizing the leading eigenvector of $\Sigma$, denoted as $v$ to define the weights of the ensemble. The log-likelihood predictions from each model are multiplied by the elements of $v$. Specifically, the spectral ensemble is defined as

$$\bar{S}(x) = \sum_{\ell=1}^{N_{perm}} S_l(x)v[\ell]. \tag{6}$$

The intuition is that if we assume the estimation errors of different estimators are independent, then the off-diagonal elements of $\Sigma$ should be approximately rank-one [Jaffe et al., 2015]. Section 4.4 demonstrates that the spectral ensemble works relatively well even for small values of $N_{perm}$. To the best of our knowledge, this is the first extension of the spectral ensemble [Jaffe et al., 2015] to anomaly detection.

# 4 EXPERIMENTS

All experiments were conducted using three different seeds. Each seed consists of three ensemble models with different random feature permutations. We used a learning rate of 1e-4 and a dropout of 0.1 for all datasets. Batch size is relative to the dataset size $N/10$ and has minimum and maximum values of 16 and 8096, respectively. Experiments were run on an NVIDIA A100 GPU with 80GB of memory.

## 4.1 SYNTHETIC EVALUATION

First, we use synthetic data to demonstrate the advantage of our variance regularization for anomaly detection. We generate simple two-dimensional data following [Buitinck et al., 2013]. The normal data is generated by drawing 300 samples from three Gaussians with a standard deviation of 1 and means on $(0, 0)$, $(-5, -5)$, and $(5, 5)$. We then generate anomalies by drawing 40 samples from two skewed Gaussians centered at $(-5, 5)$ and $(5, -5)$. We train our proposed autoregressive density estimator based on 150 randomly selected normal samples with and without the proposed variance regularizer (see Eq. 1). In Figure 3, we present the scaled log-likelihood obtained by both models. As indicated by this figure, without regularization, the log-likelihood estimate tends to attain high values in a small vicinity surrounding normal points observed during training. In contrast, the regularized model learns a distribution with

lower variance and more uniform distribution around normal points. In this example, the average area under the curve (AUC) of the Receiver Operating Characteristics curve over 5 runs of the regularized model is 98.3, while for the unregularized model, it is 79.8. This example sheds some light on the potential benefit of our regularization for anomaly detection. The following section provides more empirical real-world evidence supporting this claim.

## 4.2 REAL DATA

Experiments were conducted on various tabular datasets widely used for anomaly detection. These include 47 datasets from the recently proposed Anomaly Detection Benchmark [Han et al., 2022] and five datasets from [Rayana, 2016, Pang et al., 2019]. The datasets exhibit variability in sample size (80-619,326 samples), the number of features (3-1,555), and the portion of anomalies (from 0.03% to 39.91%). We evaluate all models using the well-known area under the curve (AUC) of the Receiver Operating Characteristics curve. We follow the same data splitting scheme as in ICL [Bergman and Hoshen, 2020, Shenkar and Wolf, 2022, Qiu et al., 2021], where the anomalous data is not seen during training. The normal samples are split 50/50 between training and testing sets.

**Baseline methods** We compare our method to density based methods like HBOS [Goldstein and Dengel, 2012], COPOD [Li et al., 2020], and [Li et al., 2022], geometric methods such as $k$-NN [Angiulli and Pizzuti, 2002], and IForest [Liu et al., 2008], and recent neural network based methods like ICL [Shenkar and Wolf, 2022], NTL [Qiu et al., 2021], GOAD [Bergman and Hoshen, 2020], and DTE [Livernoche et al., 2023]. Following [Shenkar and Wolf, 2022], we evaluate $k$-NN [Angiulli and Pizzuti, 2002] method with $k = 5$. For GOAD [Bergman and Hoshen, 2020], we use the KDD configuration, which specifies all of the hyperparameters, since it was found to be the best configuration in previous work [Shenkar and Wolf, 2022]. For DTE, we evaluated all of the three proposed configurations (DTE-NP, DTE-IG, DTE-C). For brevity, we presented the result for the best option obtained over the tested 52 datasets, DTE-NP. While many other methods are specifically designed for image data, to the best of our knowledge, this collection of baselines covers the most up-to-date methods for anomaly detection with tabular data.

**Results** In Figure 4(b), we present the AUC of our method and all baselines evaluated on 52 different tabular anomaly detection datasets. Our method outperforms previous state-of-the-art schemes by a large margin (both on average and median AUCs). Specifically, we obtained 86.0 and 92.4 mean and median AUC, better than the second-best method (ICL) by 1.2 and 2.2 AUC points, respectively. We also achieved an average rank of 3.3 over all datasets, which

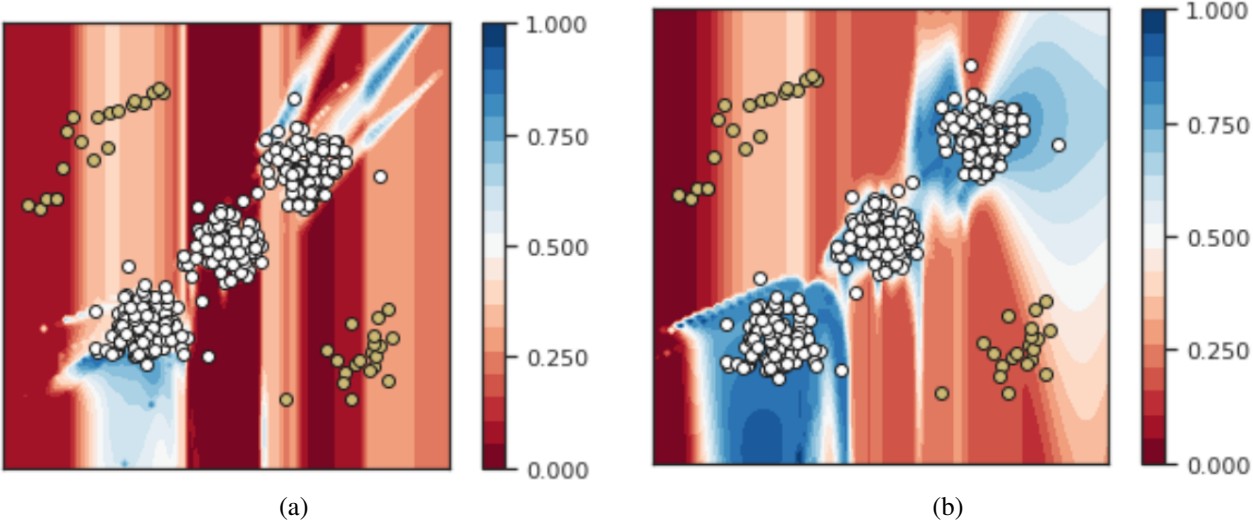

(a)                                                                                          (b)

Figure 3: Synthetic example demonstrating the effect of density stabilization. White dots represent normal samples $x_n \in X_N$, while yellow represents anomalies $x_a \in X_A$. (a): scaled unregularized log-likelihood estimation. (b): the proposed scaled regularized log-likelihood estimate. Using the proposed stabilized density estimate (right) improved the AUC of anomaly detection from 79.8 to 98.3 in this example.

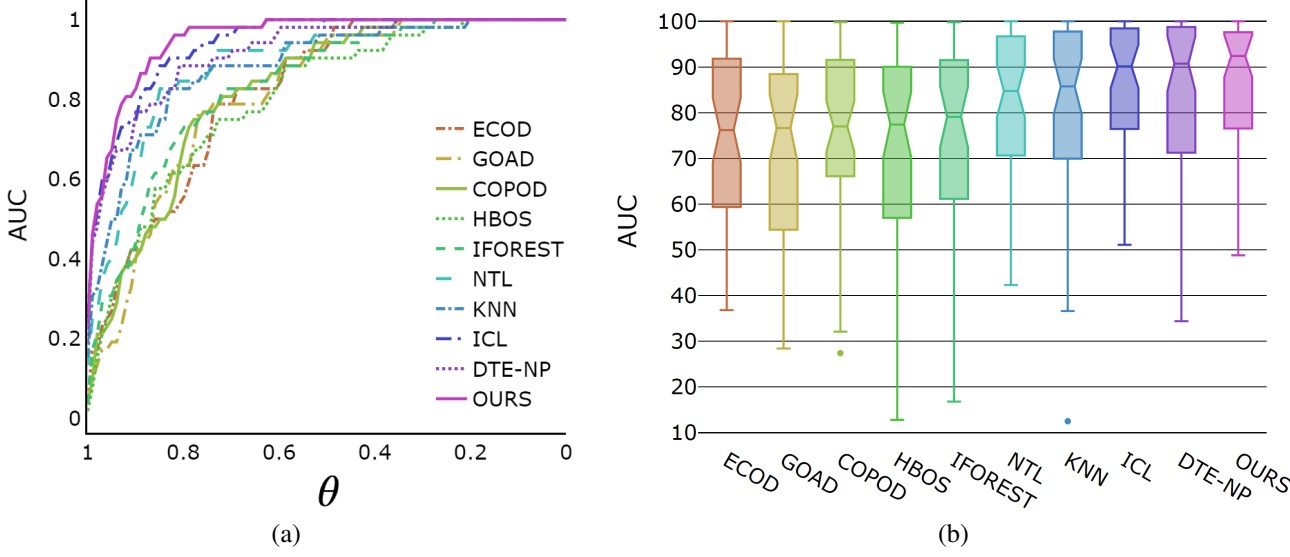

(a)                                                                                          (b)

Figure 4: (a) A Dolan-More performance profile [Dolan and Moré, 2002] comparing AUC scores of 8 algorithms applied to 52 tabular datasets. For each method and each value of $\theta$ ($x$-axis), we calculate the ratio of datasets on which the method performs better or equal to $\theta$ multiplied by the best AUC for the corresponding dataset. Specifically, for a specific method we calculate $\frac{1}{N_{data}} \sum_j \text{AUC}_j \geq \theta \cdot \text{AUC}_j^{best}$, where $\text{AUC}_j^{best}$ is the best AUC for dataset $j$ and $N_{data}$ is the number of datasets. The ideal algorithm would achieve the best score on all datasets and thus reach the left top corner of the plot for $\theta = 1$. Our algorithm yields better results than all baselines, surpassing ICL on values between $\theta = 0.95$ and $\theta = 0.82$. Furthermore, our method covers all datasets (ratio equals 1) for $\theta = 0.82$ and outperforms the second best, ICL [Shenkar and Wolf, 2022], which achieves the same at $\theta = 0.69$. This suggests that using our method on all datasets will never be worse than the leading method by more than 18%. (b) Box plots comparing the results of all methods on the 52 evaluated datasets. Each box presents the mean (red) and median (black) as well as other statistics (Q1, Q3, etc.).

surpasses the second- and third-best, 3.8 and 3.9, respectively, by ICL and DTE-NP. Furthermore, our method was

never ranked last on any of the evaluated datasets. These results indicate that our method is stable compared to the

other methods tested. The complete results are available in Table 5. We perform another performance analysis using Dolan-More performance profiles [Dolan and Moré, 2002] on AUC scores. Based on the curve presented in Figure 4, our method performs best on a larger portion of datasets for any $\theta$. $\theta \in [0, 1]$ is a scalar factor multiplying AUC obtained by the best method. For example, on all datasets, our method is never worse than $0.82$ times the highest AUC obtained by any scheme (as indicated by the intersection of our curve with the line $y = 1$). The Dolan-More curve is further explained in the caption of this figure.

Table 1: Ablation study. We evaluate the removal of several components of our model. Namely, $\lambda = 0$ indicates the removal of the stability-inducing regularizer (Eq. 1), $\Pi_\ell = I$ corresponds to no feature permutation, and mean ensemble replaces the proposed spectral ensemble by a simple mean of the different density estimators.

| Variant | $\lambda = 0$ | $\Pi_\ell = I$ | Mean Ensemble | Ours |
|---|---|---|---|---|
| Abalone | 95.6 (+1.9) | 85.7 (-8.0) | 90.6 (-3.1) | 93.7 ±0.7 |
| Annthyroid | 88.2 (-6.1) | 87.1 (-7.1) | 92.0 (-2.3) | 94.3 ±0.5 |
| Arrhythmia | 77.3 (-1.3) | 78.6 | 78.2 (-0.4) | 78.6 ±0.2 |
| Breastw | 94.1 (-5.2) | 99.2 (-0.1) | 98.5 (-0.8) | 99.3 ±0.1 |
| Cardio | 59.6 (-34.1) | 92.0 (-1.7) | 92.1 (-1.6) | 93.7 ±0.3 |
| Ecoli | 89.0 (-2.9) | 87.0 (-4.9) | 90.4 (-1.5) | 91.9 ±1.5 |
| Cover | 58.9 (-39.1) | 98.8 (-0.2) | 97.6 (-1.4) | 99.0±0.2 |
| Glass | 77.0 (-11.4) | 89.0 (+0.6) | 87.9 (-0.5) | 88.4 ±1.2 |
| Ionosphere | 96.2 (-0.2) | 96.4 | 96.0 (-0.4) | 96.4 ±0.2 |
| Letter | 71.4 (-23.8) | 94.2 (-1.0) | 93.6 (-1.6) | 95.2 ±0.3 |
| Lympho | 99.8 (+0.1) | 99.9 (+0.2) | 99.2 (-0.5) | 99.7 ±0.1 |
| Mammo. | 87.0 (-0.9) | 88.0 (+0.1) | 86.5 (-1.4) | 87.9 ±0.4 |
| Musk | 99.7 (-0.3) | 100.0 | 100.0 | 100.0 ±0.0 |
| Optdigits | 66.3 (-20.7) | 88.4 (+1.4) | 85.5 (-1.5) | 87.0 ±0.3 |
| Pendigits | 69.2 (-30.7) | 99.7 | 99.4 (-0.3) | 99.7 ±0.0 |
| Pima | 70.5 (+2.3) | 64.8 (-3.4) | 65.9 (-2.3) | 68.2 ±0.4 |
| Satellite | 68.1 (-15.2) | 84.3 (+1.0) | 82.9 (-0.4) | 83.3 ±0.2 |
| Satimage-2 | 73.3 (-26.2) | 99.0 (-0.5) | 99.3 (-0.2) | 99.5 ±0.1 |
| Shuttle | 99.6 (+0.1) | 99.0 (-0.5) | 99.5 | 99.5 ±0.2 |
| Thyroid | 97.4 (+2.0) | 94.5 (-0.9) | 93.0 (-2.4) | 95.4 ±0.1 |
| Vertebral | 52.8 (-6.0) | 52.9 (-5.9) | 56.4 (-2.4) | 58.8 ±2.1 |
| Vowels | 72.1 (-26.9) | 97.8 (-1.2) | 98.1 (-0.9) | 99.0 ±0.1 |
| Wbc | 76.7 (-19.6) | 95.8 (-0.5) | 94.6 (-1.7) | 96.3 ±0.1 |
| Wine | 93.2 (-0.1) | 94.1 (+0.8) | 90.6 (-2.7) | 93.3 ±0.5 |
| Mean | 79.4 (-10.6) | 88.7 (-0.3) | 88.8 (-0.2) | **90.0** |

## 4.3 ABLATION STUDY

We conduct an ablation study to evaluate all components of the proposed scheme.

**Variance stabilization** In the first ablation, we evaluate the properties of the proposed variance stabilization loss (see Eq. 1). We conduct an ablation with 24 datasets and compare the AUC of our model to a version that does not include the new regularization. As indicated by Table 1, there is a significant performance drop once the regularizer is removed; specifically, the average AUC drops by more

than 10 points.

**Ensemble of feature permutation** We conduct an additional experiment with the same 24 datasets to evaluate the importance of our permutation-based spectral ensemble. We compare the proposed approach to a variant that relies on a simple mean ensemble and to a variant that relies on a spectral ensemble with no feature permutation. The results presented in Table 1 demonstrate that the feature permutations and spectral ensemble help learn a reliable density estimate for anomaly detection.

## 4.4 STABILITY ANALYSIS

Here, we evaluate the stability of our approach to different values of $\lambda$, different numbers of feature permutations $N_{perm}$, for contamination in the training data, and for different types of synthetic anomalies injected to real data.

**Regularization parameter** To demonstrate that our method is relatively stable to choice of $\lambda$. We apply our framework to multiple datasets, with values of $\lambda$ in the range of $[0, 10]$. As indicated by the heatmap presented in Figure 5, adding the regularization helps improve the AUC in most datasets. Moreover, our performance is relatively stable in the range $[1, 10]$; we use $\lambda = 3.33$ in our experiments, which worked well across many datasets.

**Contaminated training data** In the following experiment, we evaluate the stability of our model to contamination in the training data. Namely, we introduce anomalous samples to the training data and evaluate their influence on our model. In Table 2, we present the AUC of our model for several datasets with different levels of training set contamination. We focus on datasets with relatively many anomalies. As indicated by these results, the performance of our model is relatively stable to anomalies in the training set.

Table 2: AUC results for various amounts of anomalies in the training data using different AD datasets. These results show that our method is relatively stable to contamination in the training set.

| Anomaly % | 1% | 3% | 5% | 0% |
|---|---|---|---|---|
| Breastw | 98.4 (-0.5) | 98.7 (-0.2) | 98.7 (-0.2) | 98.9 ±0.1 |
| Cardio | 95.1 (+0.9) | 94.3 (+0.1) | 94.6 (+0.4) | 94.2 ±0.7 |
| Pima | 67.7 (-0.4) | 67.2 (-0.9) | 67.6 (-0.5) | 68.1 ±0.8 |
| Ionosphere | 95.9 (-0.1) | 94.8 (-1.2) | 94.1 (-1.9) | 96.0±0.1 |
| Vertebral | 55.1 (+2.8) | 53.2 (+0.9) | 55.5 (+3.2) | 52.3 ±0.8 |

**Anomaly type analysis** To better understand the strengths and weaknesses of our model, we analyze its ability to detect anomalies of different types. Toward this goal, we use the semi-synthetic data created by Han et al. [2022], which includes four types of common synthetic anomalies (Local,

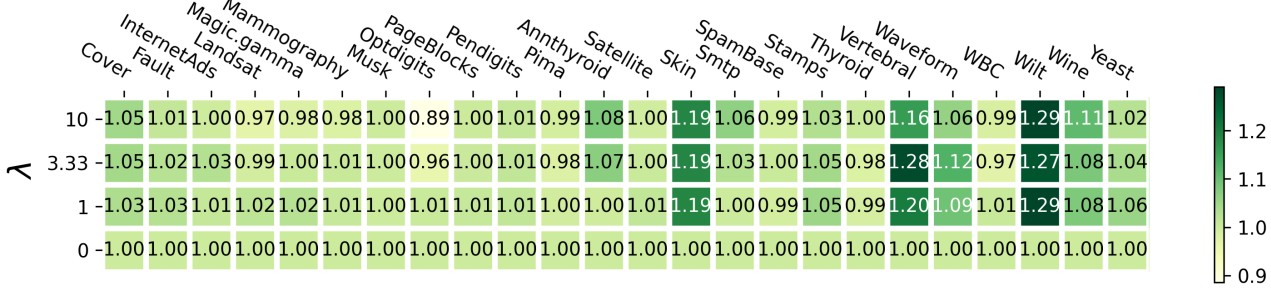

Figure 5: Stability analysis for the regularization parameter $\lambda$ balances the likelihood and the variance loss. $\lambda = 0$ indicates that no variance loss is applied. The numbers present the ratio between the AUC and the AUC obtained without regularization ($\lambda = 0$). This heatmap indicates the advantage of the proposed regularization for anomaly detection. Furthermore, observe the stability of the AUC for different values of $\lambda$.

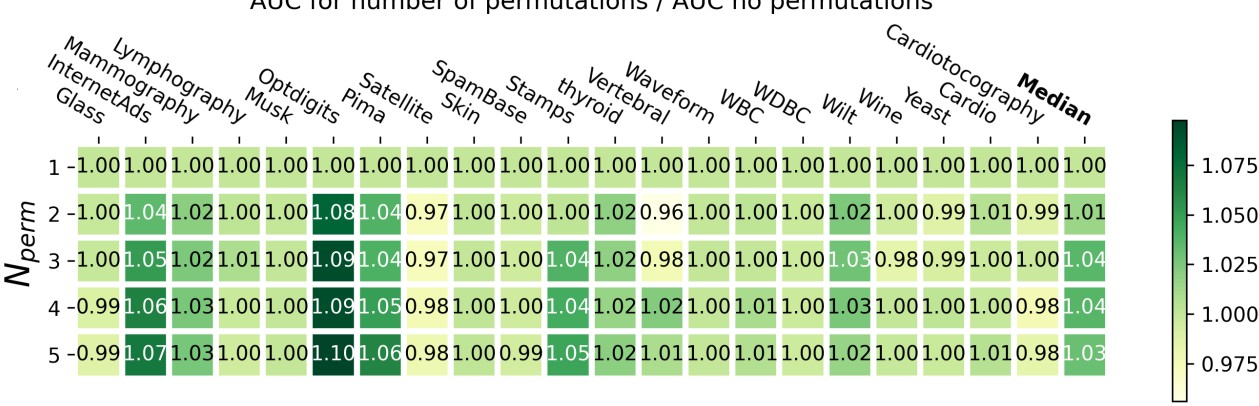

Figure 6: Stability analysis of the number of permutations. $N_{perm} = 1$ indicates that no permutations are applied, while $N_{perm} = 5$ is the result of a spectral ensemble of 5 permutations. The numbers present the ratio between the AUC of a single model and the ensemble of $N_{perm}$ permuted estimators.

Global, Dependent, and Clustered anomalies) injected into real datasets. We use the same protocol as in the original paper. The protocol discards the original datasets' anomalies as their types are unknown. Then, we generate synthetic anomalies to replace the discarded ones, maintaining the original datasets' anomaly ratio. In Figure 7, we draw box plots of our model's AUC results when applied to all datasets with the four different anomaly types. Overall, we have seen encouraging AUC results of 98.7 and 92.9 for Global and Cluster anomalies, respectively. On Local and Dependent, we have slightly worse results of 85.0 and 83.5, respectively. Intuitively, Global anomalies are generated from a distribution completely independent of the normal samples. Therefore, we expect our model to perform well on these types of anomalies. In contrast, local anomalies are generated from a scaled version of a Gaussian mixture fitted to the normal samples. Therefore, anomalies may fall in a higher likelihood function region, making them harder to identify. Raw AUC results are also available in Table 4.

**Feature permutation** To evaluate the influence of the number of feature permutations on the performance of our spectral ensemble, we run our model on several datasets with values of $N_{perm} = \{1, 2, 3, 4, 5\}$. In Figure 6, we present the AUC of our ensemble for $N_{perm} > 1$ relative to the performance of a single model, with no ensemble ($N_{perm} = 1$). This heatmap indicates that our ensemble improves performance, and $N_{perm} = 3$ is sufficient to obtain a robust spectral ensemble. Therefore, we use $N_{perm} = 3$ across our experimental evaluation. Furthermore, for the spectral ensemble, we use the absolute value of $v$, to remove arbitrary signs from this eigenvector (Eq. 6).

## 5 DISCUSSION

We revisit the problem of density-based anomaly detection in tabular data. Our key observation is that the density function is more *stable* (with lower variance) around normal samples than anomalies. We empirically corroborate our

Table 3: Robustness analysis to different types of anomalies, as detailed in Han et al. [2022]. Each column shows the AUC score of our method on various datasets used in this paper.

| Dataset | Global | Cluster | Local | Dependency |
|---|---|---|---|---|
| Vertebral | 94.6 | 93.7 | 77.0 | 87.0 |
| WDBC | 100.0 | 85.1 | 93.9 | 98.9 |
| Stamps | 99.1 | 88.4 | 85.1 | 82.7 |
| Hepatitis | 99.8 | 93.8 | 91.0 | 89.1 |
| Wine | 98.9 | 68.8 | 90.9 | 94.4 |
| Lymphography | 100.0 | 98.5 | 90.2 | 90.4 |
| Pima | 96.2 | 81.1 | 87.5 | 80.7 |
| Glass | 99.0 | 72.4 | 73.3 | 78.9 |
| Breastw | 98.2 | 94.8 | 72.0 | 75.6 |
| WPBC | 100.0 | 96.7 | 92.5 | 93.6 |
| WBC | 98.9 | 82.5 | 87.2 | 80.9 |
| Vowels | 98.4 | 87.0 | 88.7 | 90.3 |
| Yeast | 98.7 | 46.2 | 77.7 | 80.7 |
| Letter | 100.0 | 99.7 | 95.2 | 99.2 |
| Cardio | 100.0 | 94.7 | 90.3 | 92.3 |
| Fault | 100.0 | 98.7 | 84.5 | 98.2 |
| Cardiotocography | 99.9 | 95.9 | 90.1 | 84.3 |
| Musk | 100.0 | 100.0 | 99.7 | 93.7 |
| Waveform | 99.9 | 100.0 | 96.6 | 90.1 |
| Speech | 100.0 | 95.3 | 100.0 | 100.0 |
| Thyroid | 99.2 | 97.9 | 74.0 | 75.1 |
| Wilt | 83.6 | 94.7 | 70.7 | 63.9 |
| Optdigits | 100.0 | 99.9 | 98.2 | 88.9 |
| PageBlocks | 99.9 | 95.0 | 68.8 | 62.1 |
| Satimage-2 | 99.9 | 95.2 | 88.0 | 99.6 |
| Satellite | 100.0 | 99.5 | 84.9 | 99.5 |
| Landsat | 99.9 | 97.4 | 75.6 | 99.5 |
| Pendigits | 98.7 | 97.9 | 91.0 | 90.3 |
| Annthyroid | 99.4 | 99.1 | 77.2 | 73.5 |
| Mnist | 100.0 | 100.0 | 99.6 | 84.8 |
| Magic.gamma | 98.3 | 95.6 | 76.2 | 77.0 |
| Cover | 98.5 | 97.3 | 84.3 | 89.1 |
| Donors | 99.4 | 99.5 | 73.1 | 64.5 |
| Backdoor | 100.0 | 96.0 | 100.0 | 82.9 |
| Shuttle | 99.8 | 92.8 | 90.1 | 80.6 |
| Celeba | 100.0 | 89.2 | 96.7 | 73.1 |
| Fraud | 100.0 | 100.0 | 87.6 | 97.5 |
| Http | 99.7 | 99.9 | 69.1 | 55.8 |
| Skin | 92.0 | 91.7 | 57.5 | 58.5 |
| Mammography | 98.6 | 97.8 | 69.8 | 82.1 |
| Smtp | 98.7 | 100.0 | 88.2 | 45.5 |
| Mean | 98.7 | 92.9 | 85.0 | 83.5 |

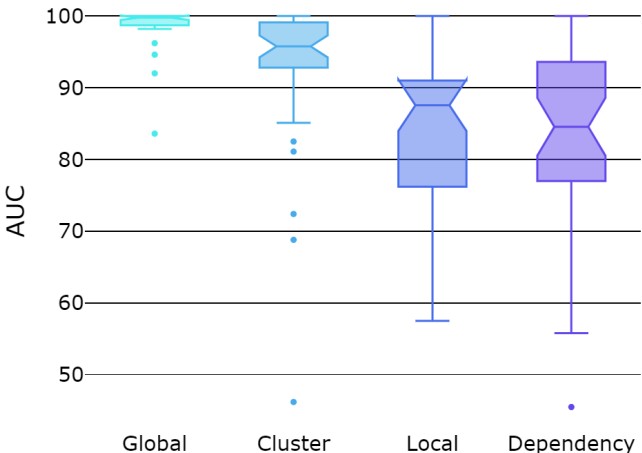

Figure 7: We evaluated the performance of our anomaly detection method on four common types of synthetic anomalies (Local, Global, Dependent, and Clustered anomalies) injected into real datasets, following the approach suggested by Han et al. [2022]. Our evaluation includes AUC results on all 52 datasets with these common anomaly types.

validate the importance of each component of our method. Finally, we present a stability analysis demonstrating that our model is relatively stable to different parameter choices and contamination in the training data.

Our work focuses on tabular datasets and does not explore other potential domains like image data or temporal signals; extending our models to these is compelling and can be performed by introducing convolution or recurrent blocks into our PNN. Our spectral ensemble adapts the supervised ensemble [Jaffe et al., 2015] via an aggregation of density functions. While the ensemble demonstrated robust empirical results, it still lacks theoretical guarantees; we believe that studying its properties is an exciting question for future work.

Another interesting question for future work, is studying the relation between our objective and the Watanabe-Akaike Information Criterion (WAIC) [Watanabe and Opper, 2010]. The WAIC is a criterion that can be used to identify out-of-distribution samples [Choi et al., 2018], and is defined as $\mathbb{E}_\theta\big[\log \hat{p}_\theta(x)\big] - \mathrm{Var}_\theta\big[\log \hat{p}_\theta(x)\big]$, which resembles our objective but is estimated by training several models each initialized with a random $\theta$.

Finally, there are several challenging datasets on which our model is still far from obtaining state-of-the-art AUC values; understanding how to bridge this gap is an open question. In Appendix D, we highlight some of these examples and analyze the relationship between the AUC of our model and the different properties of the data.

*stability* assumption using 52 publicly available datasets. Then, we formulate a modified density estimation problem that balances maximizing the likelihood and minimizing the density variance. We introduce a new spectral ensemble of probabilistic normalized networks to find a robust solution, each computed based on a different feature permutation. We perform an extensive benchmark demonstrating that our method pushes the performance boundary of anomaly detection with tabular data. We then conduct an ablation study to

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

# Supplementary Material

**Amit Rozner**[1*]     **Barak Battash**[1*]     **Henry Li**[2]     **Lior Wolf**[3]     **Ofir Lindenbaum**[1]

[1]Faculty of Engineering, Bar Ilan University, Ramat-Gan, Israel
[2]Department of Applied Mathematics, Yale University, New Haven, Connecticut
[3]School of Computer Science, Tel Aviv University, Tel-Aviv, Israel
[*]These authors contributed equally

## A   TYPES OF ANOMALIES

In this section, we detail the AUC results of our method for four different types of common synthetic anomalies (Local, Global, Dependent, and Clustered anomalies). We follow the same protocol as detailed in Han et al. [2022]. For more details please see our "Anomaly type analysis" section in our main text 4.4. In Table 4, we present all AUC results across all datasets evaluated in this work. As indicated by these results, our model works well with global anomalies, and may struggle with Dependency based or Local anomalies.

Table 4: Robustness analysis to different types of anomalies, as detailed in Han et al. [2022]. Each column shows the AUC score of our method on various datasets used in this paper.

| Dataset | Global | Cluster | Local | Dependency |
|---|---|---|---|---|
| Vertebral | 94.6 | 93.7 | 77.0 | 87.0 |
| WDBC | 100.0 | 85.1 | 93.9 | 98.9 |
| Stamps | 99.1 | 88.4 | 85.1 | 82.7 |
| Hepatitis | 99.8 | 93.8 | 91.0 | 89.1 |
| Wine | 98.9 | 68.8 | 90.9 | 94.4 |
| Lymphography | 100.0 | 98.5 | 90.2 | 90.4 |
| Pima | 96.2 | 81.1 | 87.5 | 80.7 |
| Glass | 99.0 | 72.4 | 73.3 | 78.9 |
| Breastw | 98.2 | 94.8 | 72.0 | 75.6 |
| WPBC | 100.0 | 96.7 | 92.5 | 93.6 |
| WBC | 98.9 | 82.5 | 87.2 | 80.9 |
| Vowels | 98.4 | 87.0 | 88.7 | 90.3 |
| Yeast | 98.7 | 46.2 | 77.7 | 80.7 |
| Letter | 100.0 | 99.7 | 95.2 | 99.2 |
| Cardio | 100.0 | 94.7 | 90.3 | 92.3 |
| Fault | 100.0 | 98.7 | 84.5 | 98.2 |
| Cardiotocography | 99.9 | 95.9 | 90.1 | 84.3 |
| Musk | 100.0 | 100.0 | 99.7 | 93.7 |
| Waveform | 99.9 | 100.0 | 96.6 | 90.1 |
| Speech | 100.0 | 95.3 | 100.0 | 100.0 |
| Thyroid | 99.2 | 97.9 | 74.0 | 75.1 |
| Wilt | 83.6 | 94.7 | 70.7 | 63.9 |
| Optdigits | 100.0 | 99.9 | 98.2 | 88.9 |
| PageBlocks | 99.9 | 95.0 | 68.8 | 62.1 |
| Satimage-2 | 99.9 | 95.2 | 88.0 | 99.6 |
| Satellite | 100.0 | 99.5 | 84.9 | 99.5 |
| Landsat | 99.9 | 97.4 | 75.6 | 99.5 |
| Pendigits | 98.7 | 97.9 | 91.0 | 90.3 |
| Annthyroid | 99.4 | 99.1 | 77.2 | 73.5 |
| Mnist | 100.0 | 100.0 | 99.6 | 84.8 |
| Magic.gamma | 98.3 | 95.6 | 76.2 | 77.0 |
| Cover | 98.5 | 97.3 | 84.3 | 89.1 |
| Donors | 99.4 | 99.5 | 73.1 | 64.5 |
| Backdoor | 100.0 | 96.0 | 100.0 | 82.9 |
| Shuttle | 99.8 | 92.8 | 90.1 | 80.6 |
| Celeba | 100.0 | 89.2 | 96.7 | 73.1 |
| Fraud | 100.0 | 100.0 | 87.6 | 97.5 |
| Http | 99.7 | 99.9 | 69.1 | 55.8 |
| Skin | 92.0 | 91.7 | 57.5 | 58.5 |
| Mammography | 98.6 | 97.8 | 69.8 | 82.1 |
| Smtp | 98.7 | 100.0 | 88.2 | 45.5 |
| Mean | 98.7 | 92.9 | 85.0 | 83.5 |

# B  DETAILED RESULTS

In this section, we present the full AUC results for 52 datasets with their standard deviations using all methods presented in this paper.

Table 5: AUC results on 52 datasets from widely used anomaly detection benchmarks for tabular data [Han et al., 2022] and [Rayana, 2016, Pang et al., 2019].

| Method | $k$-NN 2002 | GOAD 2020 | HBOS 2012 | IForest 2008 | COPOD 2020 | ECOD 2022 | ICL 2022 | NTL 2022 | DTE-NP 2023 | Ours |
|---|---|---|---|---|---|---|---|---|---|---|
| ALOI | 51.5±0.2 | 50.2±0.2 | 52.3±0.0 | 50.8±0.4 | 49.5±0.0 | 53.1±0.0 | 54.2±0.8 | 52.0±0.0 | 51.2±0.5 | 60.5±0.3 |
| Annthyroid | 71.5±0.7 | 93.2±0.9 | 69.1±0.0 | 91.7±0.2 | 76.8±0.1 | 79.0±0.0 | 80.5 ±1.3 | 85.2±0.0 | 92.9±0.3 | 94.3±0.5 |
| Backdoor | 94.6±0.4 | 89.3±0.5 | 72.6±0.2 | 74.8±2.9 | 79.5±0.3 | 78.2±0.0 | 92.2±0.1 | 93.5±0.1 | 93.3±1.7 | 98.8±0.2 |
| Breastw | 99.6±2.1 | 97.7±0.8 | 99.6±0.6 | 99.8±1.2 | 99.8±0.3 | 99.2±0.0 | 99.1 ±0.3 | 96.3±0.3 | 99.3±0.1 | 99.3±0.1 |
| Campaign | 74.1±0.5 | 49.0±1.9 | 80.3±0.1 | 72.9±0.1 | 78.2±0.2 | 76.8±0.0 | 74.7 ±0.8 | 76.0±0.0 | 78.8±0.3 | 81.3±0.7 |
| Cardio | 90.5±5.2 | 84.6±3.0 | 81.2±1.2 | 94.2±1.0 | 93.0±0.4 | 93.3±0.0 | 92.7 ± 0.8 | 83.2±0.1 | 91.8±0.6 | 93.7±0.3 |
| Cardio. | 71.8±2.5 | 49.1±1.0 | 46.8±0.1 | 73.8±0.2 | 66.3±0.1 | 77.9±0.0 | 78.0 ±3.2 | 76.3±0.0 | 63.8±1.9 | 75.0±0.6 |
| Celeba | 63.1±2.9 | 28.4±0.8 | 76.8±1.5 | 70.5±0.7 | 75.1±0.9 | 75.7±0.0 | 80.3 ±1.5 | 68.8±0.2 | 70.4±0.4 | 71.7±5.7 |
| Census | 67.5±0.6 | 71.6±1.0 | 65.8±2.5 | 62.9±0.1 | 67.5±1.9 | 66.0±0.0 | 60.3 ±0.8 | 53.5±1.6 | 72.1±0.4 | 66.4±1.1 |
| Cover | 88.0±5.3 | 76.0±5.3 | 60.6±0.2 | 71.3±2.3 | 86.2±0.1 | 92.1±0.0 | 96.2 ±0.6 | 98.6±0.3 | 97.7±0.6 | 99.0±0.2 |
| Donors | 100.0±9.8 | 99.5±0.1 | 78.7±0.2 | 91.3±0.2 | 81.5±0.5 | 88.8±0.0 | 99.2 ± 0.8 | 85.0±0.4 | 99.3±0.3 | 95.8±2.8 |
| Fault | 58.8±0.9 | 65.4±1.6 | 53.0±0.1 | 57.6±0.4 | 49.1±0.1 | 46.5±0.0 | 78.7 ±0.7 | 58.0±0.2 | 58.6±0.7 | 78.1±0.2 |
| Fraud | 93.1±6.4 | 86.6±0.1 | 94.5±1.0 | 93.6±0.3 | 94.0±0.0 | 95.0±0.0 | 95.2 ±0.4 | 87.5±0.3 | 95.6±1.1 | 95.3±0.0 |
| Glass | 82.3±2.2 | 82.1±6.3 | 80.3±0.5 | 74.9±1.3 | 72.5±0.4 | 70.0±0.0 | 88.1 ± 5.0 | 72.5±0.2 | 89.6±3.5 | 88.4±1.2 |
| Hepatitis | 48.3±6.4 | 32.4±6.1 | 78.0±5.0 | 75.6±2.7 | 74.9±0.3 | 74.7±0.0 | 73.0 ±5.1 | 54.0±0.7 | 93.2±3.9 | 74.2±1.6 |
| Http | 99.8±0.0 | 50.4±0.1 | 99.7±1.0 | 99.0±0.1 | 98.8±0.7 | 97.9±0.0 | 99.5 ±0.0 | 100.0±0.5 | 100.0±0.0 | 99.9±0.0 |
| InternetAds | 73.7±0.9 | 66.4±3.0 | 53.1±3.9 | 45.6±14.4 | 65.9±5.5 | 43.1±0.0 | 84.1±1.4 | 76.0±2.7 | 70.0±2.2 | 86.0±0.1 |
| Ionosphere | 91.7±3.0 | 96.5±1.1 | 62.4±0.6 | 84.6±1.3 | 77.2±0.3 | 72.7±0.0 | 98.1±0.4 | 97.9±0.6 | 97.8±1.4 | 96.4±0.2 |
| Landsat | 68.4±0.8 | 58.6±1.6 | 73.2±6.3 | 60.1±0.1 | 49.3±0.9 | 36.8±0.0 | 74.9±0.4 | 66.5±2.1 | 68.2±1.8 | 70.7±0.4 |
| Letter | 36.6±2.9 | 87.6±0.9 | 35.2±1.1 | 33.0±4.1 | 40.9±0.2 | 56.7±0.0 | 92.8 ± 0.9 | 84.8±0.3 | 34.4±1.0 | 95.2±0.3 |
| Lympho | 99.5±20.5 | 59.9±14.2 | 97.9±3.7 | 99.8±1.0 | 99.3±3.0 | 100.0±0.0 | 99.5 ± 0.3 | 97.1±2.1 | 99.9±0.1 | 99.7±0.1 |
| Magic.gamma | 84.3±0.9 | 77.3±0.2 | 74.3±0.6 | 76.8±4.0 | 68±0.3 | 63.9±0.0 | 80.9±0.1 | 82.0±0.7 | 83.6±0.8 | 85.9±0.1 |
| Mammo. | 87.2±2.4 | 54.5±2.3 | 85.6±0.3 | 88.4±0.9 | 90.5±0.1 | 90.4±0.0 | 81.1±2.0 | 82.5±0.2 | 87.6±0.1 | 87.9±0.4 |
| Mnist | 93.4±0.1 | 87.7±1.0 | 74.5±0.1 | 87.2±1.3 | 77.7±0.1 | 73.6±0.0 | 98.2±0.0 | 98.0±0.0 | 94.0±0.4 | 92.9±0.0 |
| Musk | 99.7±2.9 | 100.0±0.0 | 96.4±0.0 | 90.5±0.9 | 99.7±0.0 | 95.6±0.0 | 100.0±0.0 | 100.0±0.1 | 100.0±0.0 | 100.0±0.0 |
| Optdigits | 99.5±7.9 | 93.1±1.9 | 89.2±3.6 | 81.5±1.0 | 69.3±3.2 | 59.6±0.0 | 97.5±1.5 | 84.7±0.1 | 94.3±1.7 | 87.0±0.3 |
| PageBlocks | 58.1±1.2 | 90.4±0.4 | 87.5±0.5 | 82.1±0.1 | 80.7±0.1 | 91.5±0.0 | 98.4 ±0.2 | 93.3±0.1 | 89.3±0.3 | 94.9±0.2 |
| Pendigits | 99.9±4.3 | 85.1±3.4 | 93.8±0.0 | 96.7±0.0 | 90.7±0.0 | 92.4±0.0 | 99.5±0.1 | 97.1±0.0 | 99.6±0.2 | 99.7±0.0 |
| Pima | 68.1±2.7 | 63.2±2.3 | 70.2±0.2 | 72.9±0.2 | 65.6±0.3 | 60.3±0.0 | 59.4±0.1 | 61.7±0.3 | 81.5±2.6 | 68.2±0.4 |
| Satellite | 82.2±1.1 | 78.2±0.9 | 84.5±1.0 | 77.4±0.6 | 68.3±0.3 | 58.2±0.0 | 80.6±1.7 | 82.4±0.4 | 82.1±0.7 | 83.3±0.2 |
| Satimage-2 | 99.7±0.1 | 93.2±1.7 | 96.9±0.9 | 99.4±0.7 | 97.9±0.0 | 96.6±0.0 | 99.8±0.1 | 99.8±0.2 | 99.7±0.0 | 99.5±0.1 |
| Shuttle | 99.8±0.1 | 99.9±0.0 | 98.2±0.2 | 99.7±0.9 | 99.5±0.2 | 99.3±0.0 | 100.0 ±0.0 | 99.6±0.2 | 99.9±0.0 | 99.5±0.2 |
| Skin | 91.5±0.7 | 54.1±1.6 | 75.0±0.9 | 88.4±1.3 | 53.3±0.3 | 49.0±0.0 | 92.9±5.9 | 90.6±0.5 | 98.9±0.5 | 99.8±0.0 |
| Smtp | 92.8±2.3 | 72.2±7.7 | 84.7±0.2 | 90.5±1.5 | 92.0±0.1 | 88.0±0.0 | 83.5±2.4 | 86.7±0.1 | 93.0±2.9 | 81.2±4.9 |
| SpamBase | 77.0±4.3 | 79.4±0.8 | 82.2±0.1 | 85.6±1.2 | 72.1±0.1 | 64.6±0.0 | 74.3±0.5 | 44.1±0.0 | 83.7±0.7 | 86.1±0.2 |
| Speech | 36.9±1.8 | 54.1±4.4 | 37.0±1.2 | 40.1±0.7 | 37.4±0.8 | 46.7±0.0 | 58.9 ±2.7 | 62.5±0.2 | 41.4±0.0 | 52.9±0.1 |
| Stamps | 91.4±1.7 | 72.9±4.4 | 90.9±0.2 | 91.1±0.3 | 91.1±0.0 | 86.3±0.0 | 95.0 ±0.9 | 90.9±0.0 | 97.9±0.4 | 92.9±0.3 |
| Thyroid | 95.4±13.6 | 89.2±3.0 | 98.2±0.5 | 97.9±0.4 | 92.8±1.1 | 97.8±0.0 | 98.5 ±0.1 | 98.2±0.6 | 98.6±0.0 | 95.4±0.1 |
| Vertebral | 12.5±21.5 | 49.4±4.2 | 12.8±0.6 | 16.8±1.0 | 27.4±2.5 | 41.3±0.0 | 51.1±3.2 | 59.8±5.1 | 54.3±15.5 | 58.8±2.1 |
| Vowels | 82.6±7.2 | 97.6±0.5 | 53.4±0.1 | 62.2±1.6 | 52.8±0.0 | 59.1±0.0 | 99.7±0.1 | 98.0±0.0 | 81.4±1.4 | 99.0±0.1 |
| Waveform | 78.4±0.7 | 64.5±1.6 | 68.7±1.4 | 71.4±0.3 | 72.3±1.4 | 60.4±0.0 | 82.1 ±0.9 | 79.4±2.8 | 74.5±0.0 | 67.6±0.3 |
| WBC | 93.3±5.7 | 86.6±2.9 | 95.5±0.5 | 93.9±2.2 | 95.6±0.5 | 99.8±0.0 | 95.4±1.1 | 92.8±0.3 | 99.5±0.3 | 96.3±0.1 |
| WDBC | 98.9±0.0 | 94.8±0.5 | 94.4±7.0 | 99.2±1.3 | 98.6±0.5 | 97.2±0.0 | 99.1 ±0.0 | 99.8±6.2 | 99.5±0.4 | 99.7±0.1 |
| Wilt | 75.5±2.4 | 78.4±3.4 | 34.4±0.5 | 49.6±1.5 | 32.1±0.5 | 40.5±0.0 | 62.2±3.1 | 79.3±0.0 | 62.9±5.6 | 90.2±0.7 |
| Wine | 97.5±2.6 | 86.3±9.5 | 29.6±0.1 | 49.9±0.2 | 87.8±0.0 | 74.5±0.0 | 99.5±0.6 | 99.7±0.0 | 99.4±1.0 | 93.3±0.5 |
| WPBC | 50.3±3.7 | 51.7±0.6 | 49.2±0.0 | 49.6±1.0 | 49.2±0.0 | 45.9±0.0 | 52.3±3.4 | 42.3±0.0 | 83.2±13.5 | 52.8±0.2 |
| Yeast | 44.5±2.5 | 53.7±0.8 | 43.0±5.9 | 41.6±0.7 | 38.9±0.5 | 42.5±0.0 | 53.0 ±0.4 | 53.4±0.8 | 44.6±0.3 | 48.8±0.2 |
| Abalone | 98.9±3.2 | 54.3±7.8 | 85.4±1.2 | 89.8±1.2 | 92.4±0.9 | 85.1±0.0 | 94.3 ±0.6 | 85.1±1.0 | 94.7±0.8 | 93.7±0.7 |
| Arrhythmia | 81.8±1.9 | 64.3±8.8 | 78.5±0.8 | 80.8±0.9 | 77.4±1.4 | 79.7±0.0 | 81.7 ±0.6 | 76.5±0.9 | 50.0±0.0 | 78.6±0.2 |
| Ecoli | 98.0±8.5 | 84.7±6.8 | 42.9±1.6 | 42.0±4.2 | 90.7±1.5 | 76.7±0.0 | 86.5±1.2 | 73.1±2.3 | 88.2±0.0 | 91.9±1.5 |
| Mulcross | 100.0±3.6 | 51.3±15.8 | 98.4±0.6 | 98.4±0.4 | 73.5±0.0 | 91.4±0.0 | 100.0±0.0 | 90.5±0.0 | 100.0±0.0 | 99.9±0.0 |
| Seismic | 82.7±18.3 | 67.9±1.2 | 64.8±0.5 | 59.9±0.6 | 73.8±0.9 | 67.6±0.0 | 62.9±1.0 | 43.9±0.1 | 50.0±0.0 | 73.6±0.5 |
| Mean | 80.3 | 73.2 | 72.7 | 75.6 | 74.7 | 74.0 | 84.8 | 80.6 | 83.2 | **86.0** |
| Median | 85.8 | 76.7 | 77.4 | 79.1 | 77.0 | 76.2 | 90.2 | 84.8 | 90.7 | **92.4** |

# C DATA PROPERTIES

All datasets used in our paper were collected from widely used benchmarks for anomaly detection with tabular data. Most datasets were collected by Han et al. [2022] and appear in ADBench: Anomaly Detection Benchmark. This benchmark includes a collection of datasets previously used by many authors for evaluating anomaly detection methods. We focus on the 47 classic tabular datasets from [Han et al., 2022] and do not include their newly added vision and NLP datasets. The datasets that can be downloaded from [1] and were collected from diverse domains, including audio and language processing (e.g., speech recognition), biomedicine (e.g., disease diagnosis), image processing (e.g., object identification), finance (e.g., financial fraud detection), and more. We added five classic tabular datasets used in several recent studies, including [Rayana, 2016, Pang et al., 2019, Shenkar and Wolf, 2022]. The properties of the datasets are diverse, with sample size in the range 80-619,326, the number of features varies between 3-1,555, and the portion of anomalies from 0.03% to 39.91%. The complete list of datasets with properties appears in Table 6. Datasets from ALOI to Yeast are from [Han et al., 2022], and datasets from Abalone to Seismic are from [Rayana, 2016].

Table 6: Properties of datasets presented in this paper.

| Dataset | # of samples | # of features | Anomalies [%] |
|---|---|---|---|
| ALOI | 49534 | 27 | 3.04 |
| Annthyroid | 7200 | 6 | 7.42 |
| Backdoor | 95328 | 193 | 2.3 |
| Breast | 682 | 9 | 34.99 |
| Campaign | 41188 | 62 | 11.3 |
| Cardio | 1830 | 21 | 9.6 |
| Cardiotocography | 2114 | 21 | 9.61 |
| Celeba | 202598 | 39 | 2.2 |
| Census | 299284 | 500 | 6.2 |
| Cover | 286048 | 10 | 0.96 |
| Donors | 619326 | 10 | 1.1 |
| Fault | 1940 | 27 | 34.67 |
| Fraud | 284806 | 29 | 0.2 |
| Glass | 214 | 7 | 4.21 |
| Hepatitis | 80 | 19 | 16.2 |
| Http | 567498 | 3 | 0.39 |
| InternetAds | 1966 | 1555 | 18.72 |
| Ionosphere | 350 | 32 | 35.9 |
| Landsat | 6435 | 36 | 20.71 |
| Letter | 1600 | 32 | 6.25 |
| Lympho | 148 | 18 | 4.1 |
| Magic.gamma | 19020 | 10 | 35.16 |
| Mammography | 11182 | 6 | 2.3 |
| Mnist | 7602 | 100 | 9.21 |
| Musk | 3062 | 166 | 3.1 |
| Optdigits | 5216 | 64 | 2.81 |
| PageBlocks | 5392 | 10 | 9.46 |
| Pendigits | 6870 | 16 | 2.2 |
| Pima | 768 | 8 | 34.9 |
| Satellite | 6434 | 36 | 31.64 |
| Satimage-2 | 5802 | 36 | 1.22 |
| Shuttle | 49096 | 9 | 7.1 |
| Skin | 245056 | 3 | 20.75 |
| Smtp | 95156 | 3 | 0.03 |
| SpamBase | 4207 | 57 | 39.91 |
| Speech | 3686 | 400 | 1.65 |
| Stamps | 340 | 9 | 9.1 |
| Thyroid | 3772 | 6 | 2.1 |
| Vertebral | 240 | 6 | 12.5 |
| Vowels | 1456 | 12 | 3.43 |
| Waveform | 3442 | 21 | 2.9 |
| WBC | 222 | 9 | 4.5 |
| WDBC | 366 | 30 | 2.72 |
| Wilt | 4819 | 5 | 5.33 |
| Wine | 128 | 13 | 7.7 |
| WPBC | 198 | 33 | 23.74 |
| Yeast | 1364 | 8 | 34.16 |
| Abalone | 4177 | 8 | 6 |
| Arrhythmia | 452 | 274 | 15 |
| Ecoli | 1831 | 21 | 2.7 |
| Mulcross | 262144 | 4 | 10 |
| Seismic | 2584 | 11 | 6.5 |

---

[1]https://github.com/Minqi824/ADBench/tree/main/adbench/datasets/Classical

# D  PERFORMANCE ANALYSIS

In this section, we evaluate the relationship between different data properties and the performance of our model. First, we present scatter plots of the AUC of our model vs. the portion of outliers, number of features, and number of samples in each data. All these scatter plots are presented in Fig. 8. We further present the rank of our method as the color of each marker (dataset) in the scatter plot. To analyze these results, we computed correlation values of -0.27, -0.12, and 0.18, indicating the relation between the AUC and the portion of outliers, the number of features, and the number of samples, respectively. Since these are considered weak correlations, it is hard to deduce from these values what regime is best or worst for our algorithm.

Datasets on which the proposed approach was ranked 7.5, 8, and 9 include Shuttle, Waveform, and Smtp, respectively. On Shuttle, we obtain an AUC of 99.5; therefore, we do not consider this to be a performance gap. On Waveform and Smtp, our algorithm was surpassed by 10-20 %. Since these datasets have a large variance ratio $\sigma_a^2/\sigma_n^2 > 1$, we suspect a stronger regularization could improve performance. This is also evident in these datasets' performance variability demonstrated in Fig. 5 when varying $\lambda$.

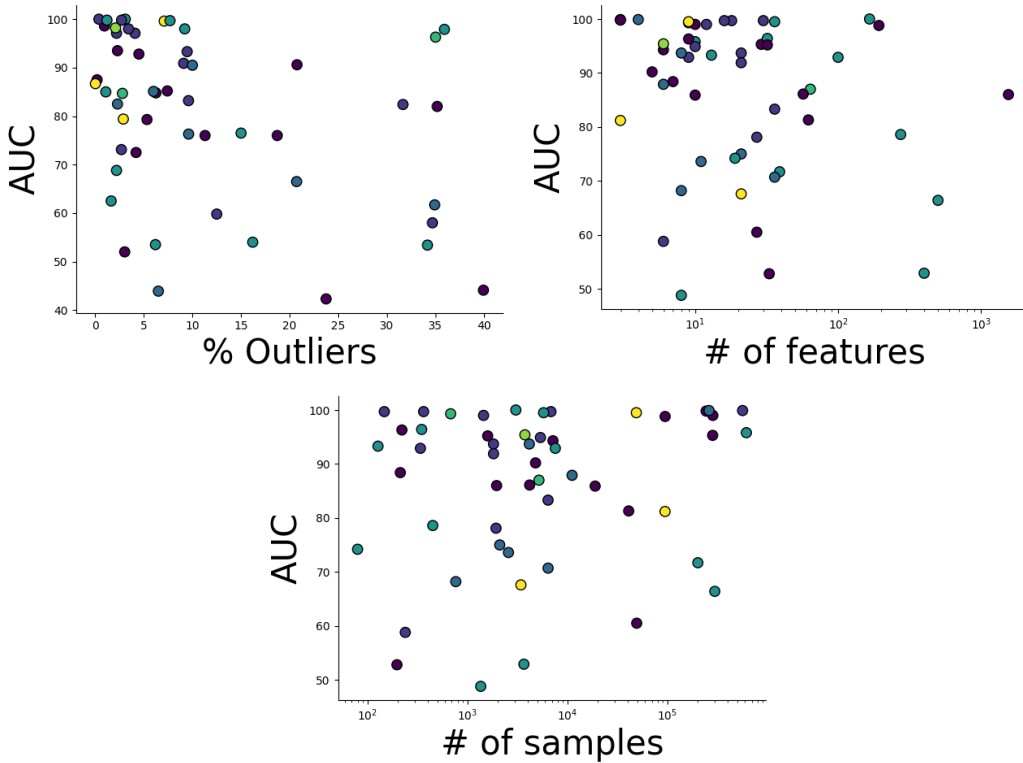

Figure 8: Scatter plots comparing the AUC of our model and different properties of the datasets, including % of outliers, # of features, and # of samples. Each dot represents a dataset, the $y$-axis represents the AUC, and the color indicates the rank of our method for the specific dataset.