# OpenReview forum: "Anomaly Detection with Variance Stabilized Density Estimation"
_auai.org/UAI/2024/Conference — UAI 2024 poster_

### Official Review · Reviewer_Ekja · 2024-03-11

**Q2-1 Originality-Novelty:** 3
**Q2-2 Correctness-Technical Quality:** 3
**Q2-5 Clarity Of Writing:** 4

**Q1 Summary And Contributions:**

A new algorithm for multivariate anomaly detection. The algorithm makes use of probability theory. It is based on the assumption that density function is stable around inlier points. The proposed method minimizes the variance around inliers.

**Q2-3 Extent To Which Claims Are Supported By Evidence:**

3: Good: the main claims are supported by convincing evidence (in the form of adequate experimental evaluation, proofs, (pseudo-)code, references, assumptions).

**Q2-4 Reproducibility:**

3: Good: key resources (e.g. proofs, code, data) are available and key details (e.g. proofs, experimental setup) are sufficiently well-described for competent researchers to confidently reproduce the main results.

**Q3 Main Strengths:**

The simplicity and elegance of the method.
The procedure is based on an intuitive idea, which makes sense to be checked-experimented.

I give value to the experimentation on the synthetic dataset, which can give more clues than extensive experimentations on real datasets. I propose to the authors to extend this experimentation on synthetic datasets, with more complex patterns. The experimented one gives information about the behaviour of the algorithm, but it is quite simplistic.

Extensive experimentation on real datasets, and large comparison with state of the art anomaly detection algorithms. As the authors do a "multiple comparison - multiple algorithms compared in multiple datasets", I propose to use the following methodology for assessing the significance of the difference of the results:
https://jmlr.org/papers/v7/demsar06a.html

**Q4 Main Weakness:**

Do the authors consider that there can be datasets where anomalies do not fullfill the exposed assumption of the method ("i.e. low variance around inliers")? Is it possible to "pre-check" it in the dataset, prior to using (or not) the proposed method, assuming that its behaviour would be poor in case it is not fullfilles? I would appreciate if the authors discuss about this issue, i.e. datasets where the assumption of the method is not fullfilled. It would be interesting to run the proposed algorithm in a synthetic domain where anomalies do not fullfill this assumption of "large variance around outliers".

Equation 2 for likelihood estimation -- the order of the features in the conditioning part. This seems a weak point of the proposed algorithm. While the authors are aware about the "choice" of the features' order, different permutation of features orders are created and thus, finally having a "feature permutation ensemble" to do the final density estimation. I would like the authors discuss more in depth about this "suppossed" weakness of their proposal.

In their "future work" section the authors also mention the "lack of theoretical guarantees" of the proposal. While intuitive, a formal study about their behaviour, maybe in synthetic domains, will enrich the paper.

I miss a computational order analysis of the algorithmic pipeline.

**Q5 Detailed Comments To The Authors:**

Please check previous Q3 and Q4

**Q9 Complying With Reviewing Instructions:**

Yes

---

> ### Author Rebuttal · Authors · 2024-04-07
>
> We thank the reviewer for pointing out the simplicity and elegance of our method and experimental results. Below, we address all the points raised by the reviewer.
>
> **W1. Synthetic datasets:** Thank you for your valuable feedback regarding the use of more synthetic datasets. We would like to inform you that this is already a part of our roadmap. We have initiated the process of exploring various types of realistic synthetic anomalies, which can be observed in Table 2. Additionally, we plan to evaluate:
>
> 1. Synthetic data in which our assumption holds vs. breaks.
>
> 2. Using methods for synthetic tabular data generation, as mentioned in [1], to make the synthetic environment more authentic.
>
> 3. Investigating the impact of correlated features, both in normal samples and anomalies.
>
> 4. Generating synthetic anomalies in real data [2].
>
> Thank you for your interest in our work.
>
> **Q3 Significance of difference analysis:** Thank you for your suggestion; we will add this statistical analysis to the manuscript.
>
> **W2. Algorithms for predicting performance on new data:** The reviewer has brought up an intriguing point about our regularization term. While it is meant to minimize the log-likelihood variance for normal training samples, it may not always be helpful and could even harm performance. Therefore, it would be interesting to come up with a metric or scheme that can assess whether our regularization would be beneficial or not. One possible approach to quantify this would be first to fit an unregularized density estimator to new data and then carefully analyze the statistics of datasets where our regularizer improved results compared to an unregularized model. We believe this could be a starting point for future research on this topic.
>
> **W3 “Equation 2 for likelihood estimation”:** We chose to use the probabilistic neural network (PNN) [3] - an autoregressive model - because it has demonstrated superior performance in numerous density estimation benchmarks, as compared to other models, such as Autoregressive Flows [4], Transformation Autoregressive Networks [5], and RealNVP [6]. Our new components have been shown to significantly enhance the capabilities of density-based models through extensive ablation testing. As such, we agree with the reviewer's observation that our autoregressive model that relies on random permutation may not be optimal for an accurate representation of the data density. However, since our model aims for a biased estimation of the density (due to the regularizer) and since we only care about the ability to identify anomalies, we don’t see this as a major issue. Specifically, since our model leads to SOTA results on this highly competitive task, we believe that the autoregressive nature has only a minor impact on our findings and that our components can work well with other density estimators.
>
> **Q4 Future work:** We acknowledge the reviewer's valuable suggestion that theoretical results and an empirical systematic evaluation to support our method would be of great value to the community. We are currently studying the theoretical aspects of the new regularization term (Eq.1) and the spectral ensemble (Eq. 6). Furthermore, as mentioned earlier, we are also studying the different properties of our models using synthetic data (see the response to W1 and W2 above).
>
> **Q4 “Computational order analysis”:** Thank you for bringing this up. Below, we have provided additional details about the computational complexity of our model.
>
> We define $O(density)$ as the computational complexity of the density estimator. To estimate the density, we use an ensemble of three estimators, denoted by $r=3$. We calculate the covariance of the $N \times r$ matrix, where $N $represents the number of training samples. Then, we compute the leading eigenvector for the $r \times r$ covariance matrix. This adds a complexity of $O(r log r)$. Therefore, the overall complexity of our model is $r O(density) +O(r log r)$.
>
> Regarding our autoregressive model, the complexity of the density estimator depends on the number of iterations required for convergence (k), the number of samples in the batch (B), and the number of features (D). Specifically, using stochastic gradient descent, the complexity scales as $O(kBD)$.
>
> Importantly, our inference time is incredibly low as it only requires a forward pass through the shallow neural network.
>
> We are keen to improve our paper and hope we have addressed all the reviewers' comments. We would happily respond within the remaining discussion time if there are still open issues.

---

### Official Review · Reviewer_cnBw · 2024-03-22

**Q2-1 Originality-Novelty:** 3
**Q2-2 Correctness-Technical Quality:** 3
**Q2-5 Clarity Of Writing:** 3

**Q10 Ethical Concerns:**

No.

**Q1 Summary And Contributions:**

The authors propose an anomaly detection algorithm using a regularized density estimator.  Towards this end, they introduce a novel assumption that the density of the non-anomalous samples should have low variance.  Their proposed estimator is applied to many experimental settings, lending credence to their claims.

**Q2-3 Extent To Which Claims Are Supported By Evidence:**

3: Good: the main claims are supported by convincing evidence (in the form of adequate experimental evaluation, proofs, (pseudo-)code, references, assumptions).

**Q2-4 Reproducibility:**

3: Good: key resources (e.g. proofs, code, data) are available and key details (e.g. proofs, experimental setup) are sufficiently well-described for competent researchers to confidently reproduce the main results.

**Q3 Main Strengths:**

The paper provides very strong intuition and motivation for their approach, by empirically demonstrating their assumption on a wide variety of experimental settings.  They include a very thorough literature review of related works and the paper is generally well written.  The simulations are also quite convincing.

**Q4 Main Weakness:**

None.

**Q5 Detailed Comments To The Authors:**

For the underlying density estimator, the authors propose using an autoregressive model.  However, this is very counterintuitive to me, since there is no underlying autoregressive nature.  I am wondering whether the authors have any intuition why this outperformed other models.

I am also curious about the low-variance assumption.  In the simulations, it is shown that most datasets satisfy the "stable" density assumption (Figure 2).  On the datasets where the assumption seems to not be satisfied, the corresponding AUC values tend to be, on average, a bit worse than some of the competing methods (Appendix B).  Is there something unique to those datasets that make them not satisfy your assumption and, given a new dataset, is there a way to decide whether the assumption might reasonably be satisfied?

AUC is defined in Section 4.2, but is used already in Section 4.1.  It might be helpful to define it at the first point of usage.

**Q9 Complying With Reviewing Instructions:**

Yes

---

> ### Author Rebuttal · Authors · 2024-04-05
>
> We thank the reviewer for valuing our idea and its intuition, as well as our experimental settings. Below, we address all points raised by the reviewer.
>
> **Q5 Using an autoregressive model:** We chose to use the probabilistic neural network (PNN) [1] - an autoregressive model - because it has demonstrated superior performance in numerous density estimation benchmarks, as compared to other models such as Autoregressive Flows [2], Transformation Autoregressive Networks [3], and RealNVP [4]. Our new components have been shown to significantly enhance the capabilities of density-based models through extensive ablation testing. As such, we agree with the reviewer's observation that other density-based models may also benefit from our new components and that the autoregressive nature has only a minor impact on our findings.
>
> **Q5 Predicting the model's performance on new data:** The reviewer has brought up an intriguing point about our regularization term. While it is meant to minimize the log-likelihood variance for normal training samples, it may not always be helpful and could even harm performance. Therefore, it would be interesting to come up with a metric or scheme that can assess whether our regularization would be beneficial or not. One possible approach to quantify this would be first to fit an unregularized density estimator to new data and then carefully analyze the statistics of datasets where our regularizer improved results compared to an unregularized model. We believe this could be a starting point for future research on this topic.
>
> **Q5 AUC definition:** Thanks for the comment; we will put the definition prior to the first usage.
>
> We are keen to improve our paper and hope we have addressed all the reviewers' comments. We would happily respond within the remaining discussion time if there are still open issues.
>
> **References:**
>
> [1] Li, Henry, and Yuval Kluger. "Neural inverse transform sampler." International Conference on Machine Learning. PMLR, 2022.
>
> [2] Papamakarios, George, Theo Pavlakou, and Iain Murray. "Masked autoregressive flow for density estimation." Advances in neural information processing systems 30 (2017).
>
> [3] Oliva, Junier, et al. "Transformation autoregressive networks." International Conference on Machine Learning. PMLR, 2018.
>
> [4] Dinh, Laurent, Jascha Sohl-Dickstein, and Samy Bengio. "Density estimation using real nvp." arXiv preprint arXiv:1605.08803 (2016).

---

### Official Review · Reviewer_DZYC · 2024-03-25

**Q2-1 Originality-Novelty:** 3
**Q2-2 Correctness-Technical Quality:** 3
**Q2-5 Clarity Of Writing:** 3

**Q1 Summary And Contributions:**

The paper suggests a novel anomaly detection method suitable for tabular data, which uses a modified density estimation problem focusing on areas with lower variance around normal samples. The suggested method employs a spectral ensemble of autoregressive models to learn a variance-stabilized distribution, which achieve state-of-the-art performance across the datasets without requiring extensive hyperparameter tuning. The method is empirically evaluated using 52 real-world datasets with positive results.

**Q2-3 Extent To Which Claims Are Supported By Evidence:**

3: Good: the main claims are supported by convincing evidence (in the form of adequate experimental evaluation, proofs, (pseudo-)code, references, assumptions).

**Q2-4 Reproducibility:**

4: Excellent: key resources (e.g. proofs, code, data) are available and key details (e.g. proof sketches, experimental setup) are comprehensively described for competent researchers to confidently and easily reproduce the main results.

**Q3 Main Strengths:**

The paper describes an anomaly detection method that is based upon an observation presented in prior studies that the density functions around anomalies show higher variance compared to normal data points. The implementation of the anomaly detection method is provided, and it is evaluated over 52 open datasets (also used in prior works) which gave positive results.

**Q4 Main Weakness:**

The main weakness that I can identify is the lack of theoretical guarantees regarding its validity. However, the authors point that out in the paper, and also argue that the empirical results in the present study should be seen as an indication that further research is needed.

**Q5 Detailed Comments To The Authors:**

Some minor mistakes:
* Section 2: last sentence: anomlies -> anomalies.
* Section 4.3 subsection "Ensemble of feature permutation": 25 datasets -> 24 datasets.
* Fig. 2 is a bit hard to read so maybe it is possible to decrease label fonts and increase the size of the bar plot.

**Q9 Complying With Reviewing Instructions:**

Yes

---

> ### Author Rebuttal · Authors · 2024-04-05
>
> We thank the reviewer for their feedback and constructive comments and for acknowledging the significance of our empirical results, which are based on 52 datasets and several leading baselines. Below, we address all points raised by the reviewer.
>
> **Q4 Theoretical justification:**  We appreciate the reviewer's suggestion. We agree that having theoretical results to support our method would be beneficial. Currently, we are working on follow-up studies to examine the theoretical validation for both our new regularization term (Eq.1) and the spectral ensemble (Eq. 6). As mentioned on page 8 of the main text, we have already noted this as a limitation of our research:
>
> “While the ensemble demonstrated robust empirical results, it still lacks theoretical guarantees; we believe that studying its properties is an exciting question for future work. Another interesting question for future work is the relationship between our objective and the Watanabe-Akaike Information Criterion (WAIC) [Watanabe and Opper, 2010]. The WAIC is a criterion that can be used to identify out-of-distribution samples [Choi et al., 2018], and is defined as Eθ log ˆpθ(x) − Varθ log ˆpθ(x), which resembles our objective but is estimated by training several models each initialized with a random θ.”
>
> Regardless, our work is primarily algorithmic and empiric. It brings several stand-alone contributions to the field that are important to the community, even without theoretical justifications. Specifically, we introduce (1) a new regularized density estimation problem that enhances the capabilities of density-based anomaly detection, which usually underperforms compared to one-class classification models. (2) We present a spectral ensemble of density estimators, each based on a different feature permutation. (3) We use an ablation to demonstrate that both components improve anomaly detection. (4) We conduct an extensive benchmark and show that our model achieves state-of-the-art results on tabular anomaly detention, which is a highly competitive task in machine learning.
>
> Therefore, we believe that the absence of theoretical explanations in our work should be viewed as a minor limitation that does not overshadow the independent contribution we bring to the field.
>
> **Q5 Minor mistakes and presentation:** Thanks for pointing out these typos and suggesting this idea to improve the readability of our paper. We have included these suggestions in our revised version.
>
> We are keen to improve our paper and hope we have addressed all the reviewers' comments. We would happily respond within the remaining discussion time if there are still open issues.

---

### Official Review · Reviewer_HtcA · 2024-03-25

**Q2-1 Originality-Novelty:** 2
**Q2-2 Correctness-Technical Quality:** 2
**Q2-5 Clarity Of Writing:** 4

**Q1 Summary And Contributions:**

This work studies the anomaly detection problem, particularly focusing on density-based methods. In these methods, anomalies are typically identified by their low likelihood values. The limitation is the tendency of these approaches to overfit. To address this challenge, the study introduces an approach by modifying the density estimation process. This modification stems from the empirical observation that normal data exhibits relatively lower variance compared to abnormal data. Leveraging this insight, the authors propose incorporating variance as a penalty term in the model fitting loss (log-likelihood). This adjustment aims to encourage the learned density distribution to exhibit reduced variance. Furthermore, the study introduces a spectral ensemble of autoregressive models to facilitate learning a variance-stabilized distribution.

**Q2-3 Extent To Which Claims Are Supported By Evidence:**

2: Fair: the main claims are somewhat supported by evidence (but the experimental evaluation may be weak, or does not match entirely with the claims, important baselines may be missing, proofs contain important ideas but lack rigor, algorithmic details are only discussed superficially, references are imprecise, assumptions are not sufficiently motivated or explicated, etc.).

**Q2-4 Reproducibility:**

3: Good: key resources (e.g. proofs, code, data) are available and key details (e.g. proofs, experimental setup) are sufficiently well-described for competent researchers to confidently reproduce the main results.

**Q3 Main Strengths:**

1. Anomaly detection is an important problem with a longstanding history. Contributing to this saturated field is challenging.
2. The authors spend effort investigating the real-world dataset. They gain insight through empirical investigation and apply the insight to design algorithms. I think this is impressive.
3. The paper is well-written and easy to follow.

**Q4 Main Weakness:**

1. The proposed method is based on a strong assumption var n < var a. (See Q5. 1)

2. Adding a regularization term to avoid overfitting seems not novel.

3. No theory justification.

**Q5 Detailed Comments To The Authors:**

1. the assumption var n < var a sounds strange to me. Are those tabular datasets all i.i.d datasets? I think, at least for time-dependent data, the assumption does not hold. Also, if we have this assumption, why don't we use some simple rule-based method to identify anomalies? For example, we identify an anomaly once the tabular value is out of a certain scale.

2. Regarding fitting models for tabular data, the stats of the art model are still tree ensemble models.

**Q9 Complying With Reviewing Instructions:**

Yes

---

> ### Author Rebuttal · Authors · 2024-04-05
>
> We thank the reviewer for acknowledging our comprehensive empirical investigation, writing style, and clarity of our paper. Below, we address all points raised by the reviewer.
>
> **Q1/W1. Variance assumption:** We want to clarify that the variance assumption is based not on the data but on the probability distribution. Specifically, the assumption is that var(p_n)<var(p_a) and not that var(n)<var(a), as mentioned by the reviewer. This variance assumption is used as a motivation for introducing our regularizer. Our empirical results support this assumption, demonstrating that it holds for most of the evaluated real datasets.
> It is important to note that our training procedure does not explicitly rely on the var(p_n)<var(p_a) assumption. Instead, we introduce a regularization that is motivated by this assumption to encourage low values of var(p_n) (with no restriction on var(p_a)). We want to emphasize that the assumption is never used to determine which samples are normal and which ones are not. Therefore, our regularizer can be seen as a soft implementation of this assumption. We would like to clarify that the rule-based method suggested by the reviewer does not align with our assumption since it is based on var(n)<var(a) rather than var(p_n)<var(p_a). We acknowledge that both normal samples and anomalies may have a similar range, but their density (and variance) could still differ around normal samples compared to anomalies.
>
> **Q2. Tree-based models:** We understand that the reviewer is making a reference to the state-of-the-art (SOTA) models in supervised learning with tabular data. However, our focus lies in anomaly detection without labels, which limits us to comparing our model to the closest unsupervised model that relies on forest ensembles, such as IForest. It is worth noting that IForest is a strong competitor, but it is still far from being the SOTA, as demonstrated in Figure 4. If the reviewer knows of any other unsupervised tree-based model that performs well in the anomaly detection setting, we are open to including it in our revised manuscript.
>
> **W2. Novelty:** Optimization and regularization are crucial concepts in machine learning. Regularization is a well-established technique, but its effectiveness and suitability may differ depending on the problem. Researchers have made significant progress in advancing this field by designing objective functions that exploit regularization for specific tasks. In some cases, regularization has proven to be especially useful, such as in [1,2]. However, while previous studies have shown that regularization can be beneficial for density-based anomaly detection [3,4,5], a specific type of regularization has not yet been explored for this purpose.
> Designing an appropriate regularization for improving performance in unsupervised tasks is not straightforward. Our intuition is based on the widely used assumption that normal data has a simple structure, while anomalies do not follow a clear pattern and can be caused by many unknown factors [6]. In the context of our density-based model, we translated this assumption into a regularization imposed on the variance of the density around normal samples. We conducted extensive experiments on both synthetic and real data and found that this new regularization has great potential for the field of anomaly detection. We are confident that our results could also be useful to other researchers interested in this field. Therefore, we believe that the development of this new regularization is novel and important for the community.
>
> **W3.Theoretical justification:** We appreciate the reviewer's suggestion. We agree that having theoretical results to support our method would be beneficial. Currently, we are working on follow-up studies to examine the theoretical validation for both our new regularization term (Eq.1) and the spectral ensemble (Eq. 6). As mentioned on page 8 of the main text, we have already noted this as a limitation of our research. Regardless, our work is primarily algorithmic and empiric. It brings several stand-alone contributions to the field that are important to the community, even without theoretical justifications. Specifically, we introduce (1) a new regularized density estimation problem that enhances the capabilities of density-based anomaly detection, which usually underperforms compared to one-class classification models. (2) We present a spectral ensemble of density estimators, each based on a different feature permutation. (3) We use an ablation to demonstrate that both components improve anomaly detection. (4) We conduct an extensive benchmark and show that our model achieves state-of-the-art results on tabular anomaly detention, which is a highly competitive task in machine learning. Therefore, we believe that the absence of theoretical explanations in our work should be viewed as a minor limitation that does not overshadow the independent contribution we bring to the field.

---

### Meta-Review · Area_Chair_b2qQ · 2024-04-15

Technically solid paper with strong empirical evaluation combined with clear presentation and no open issues. It is not extremely novel or insightful, but the key ideas are intuitive and easy to communicate also for a broad audience.